# Successful transmission and transcriptional deployment of a human chromosome via mouse male meiosis

Christina Ernst[1], Jeremy Pike[1], Sarah J Aitken[1,2], Hannah K Long[3,4,5], Nils Eling[1], Lovorka Stojic[1], Michelle C Ward[1], Frances Connor[1], Timothy F Rayner[1], Margus Lukk[1], Robert J Klose[3], Claudia Kutter[6], Duncan T Odom[1]*

[1]Cancer Research UK Cambridge Institute, University of Cambridge, Cambridge, United Kingdom; [2]Department of Histopathology, Addenbrooke's Hospital, Cambridge, United Kingdom; [3]Department of Biochemistry, University of Oxford, Oxford, United Kingdom; [4]Institute of Stem Cell Biology and Regenerative Medicine, Stanford University School of Medicine, Stanford, United states; [5]Department of Chemical and Systems Biology, Stanford University School of Medicine, Stanford, United States; [6]Department of Microbiology, Tumor and Cell Biology, Science for Life Laboratory, Karolinska Institute, Stockholm, Sweden

*For correspondence: Duncan. Odom@cruk.cam.ac.uk

**Abstract** Most human aneuploidies originate maternally, due in part to the presence of highly stringent checkpoints during male meiosis. Indeed, male sterility is common among aneuploid mice used to study chromosomal abnormalities, and male germline transmission of exogenous DNA has been rarely reported. Here we show that, despite aberrant testis architecture, males of the aneuploid Tc1 mouse strain produce viable sperm and transmit human chromosome 21 to create aneuploid offspring. In these offspring, we mapped transcription, transcriptional initiation, enhancer activity, non-methylated DNA, and transcription factor binding in adult tissues. Remarkably, when compared with mice derived from female passage of human chromosome 21, the chromatin condensation during spermatogenesis and the extensive epigenetic reprogramming specific to male germline transmission resulted in almost indistinguishable patterns of transcriptional deployment. Our results reveal an unexpected tolerance of aneuploidy during mammalian spermatogenesis, and the surprisingly robust ability of mouse developmental machinery to accurately deploy an exogenous chromosome, regardless of germline transmission.

## Introduction

Most human aneuploidies originate maternally. The most common viable aneuploidy in humans is Down Syndrome, which is caused by an extra copy of chromosome 21 that is maternally inherited in over 90% of all cases (*Hultén et al., 2010*). An elegant mouse model of human Down Syndrome is the aneuploid Tc1 mouse, which transmits an almost complete copy of human chromosome 21 (HsChr21) via the female germline (*O'Doherty et al., 2005*; *Sheppard et al., 2012*). More generally, passage of aneuploid DNA via the female germline is preferred in the majority of trisomic mouse models, most of which exhibit total male sterility (*Davisson et al., 2007*; *Hernandez and Fisher, 1999*). Across mammals, efficient and stable male germline transmission of foreign aneuploid DNA has only been reported in mice for comparatively small human artificial or fragmented chromosomes (*Tomizuka et al., 1997*; *Voet et al., 2001*; *Weuts et al., 2012*).

The mechanisms of mammalian gametogenesis disfavour paternally-derived aneuploidies as a result of sexual dimorphism in meiosis: female meiosis can extend over decades and is thus more

error-prone, whereas male meiosis is a continuous process with stringent quality checkpoints (*Hunt and Hassold, 2002*). Indeed, human gametes show an order of magnitude difference in the occurrence of chromosomal abnormalities between oocytes (20%) and sperm (2–4%) (*Martin et al., 1991*).

Male- and female-derived haploid genomes furthermore differ in the epigenetic reprogramming they experience during gametogenesis, fertilization, and zygotic activation (*Kota and Feil, 2010*; *Oswald et al., 2000*). Throughout this process, the maternally passaged haploid genome remains histone-bound (*Cantone and Fisher, 2013*; *Kimmins and Sassone-Corsi, 2005*). In stark contrast, spermatogenesis results in a form of DNA that is 6–20 times more compact than histone-bound DNA (*Balhorn, 2007*; *Dadoune, 1995*). This compaction is achieved by replacing histone proteins with sperm-specific protamine proteins, leaving very few histones bound to DNA (*Brykczynska et al., 2010*; *Erkek et al., 2013*; *Hammoud et al., 2009*). After gametic fusion, these protamines must be removed and replaced *de novo* with histones by the zygote's epigenetic machinery to initiate development.

Sperm development is associated with massive and genome-wide derepression of transcription (*Soumillon et al., 2013*). During male meiosis, almost all genes, as well as otherwise-silenced repeat elements, are expressed at extremely high levels (*Ward et al., 2013*). Once sperm mature, transcription is almost entirely silenced, only to be reactivated after fertilization during the maternal to zygotic transition (*Hammoud et al., 2014*). Whether this male germline-specific transcriptional activation and repression can accurately handle exogenous repeat sequences is entirely unexplored.

Here, we reveal that an almost complete copy of human chromosome 21 can be readily transmitted via mouse sperm in the Tc1 aneuploid mouse model of Down Syndrome. We demonstrate that the male mouse-transmitted human chromosome is accurately regulated and transcribed in derived somatic tissues, despite having undergone chromatin condensation and epigenetic reprogramming associated with spermatogenesis.

## Results

### Male mice carrying human chromosome 21 exhibit a subfertility phenotype

To assess the fertility of male mice carrying human chromosome 21 (Tc1), we performed phenotypic and histological comparisons with wild-type littermates that did not inherit human chromosome 21 (Tc0). Tc1 males showed significantly decreased testis size and weight, as well as markedly decreased sperm count (*Figure 1A,B*). Tc1-associated phenotypes appeared to be specific to testes, as total body weight and liver weight (as a representative somatic tissue) were indistinguishable between Tc1 and Tc0 mice (*Figure 1B*).

Histologically, Tc1 testes showed subfertility phenotypes as compared to normal testes from Tc0 littermates. The Tc1 males' subfertility phenotype was characterized by the absence (spermatogenic arrest) or reduced frequency (hypo-spermatogenesis) of secondary spermatocytes, as well as the absence of any cell types derived from these (*Figure 1C*) (*Borg et al., 2010*). Neither Tc1 nor Tc0 mice demonstrated other subfertility phenotypes, such as Sertoli cell-only syndrome, tubular sclerosis, or fibrosis (*Dohle et al., 2012*).

Based on Dohle *et al.* and Creasy *et al.*, we developed a system to grade spermatogenesis based on a seminiferous tubule scoring ranging from completely normal spermatogenesis (Grade I) to maturation arrest (no production of mature sperm: Grade VI) with two intermediate steps of hypo-spermatogenesis (mild: Grade II and severe: Grade III) (*Figure 1—figure supplement 1A*). While the vast majority of tubules in wild-type littermates showed normal spermatogenesis, we found phenotypically normal tubules marked by the production of mature sperm (*Figure 1C*, blue arrowheads) at lower frequencies than abnormal tubules that show very little or no sperm production in Tc1 males (*Figure 1—figure supplement 1B*). Interestingly, while defective tubules in wild-types mainly displayed an absence of mature sperm, defective tubules in Tc1 males often displayed failures in chromosome segregation during metaphase of meiosis I (*Figure 1C*, red arrowheads and *Figure 1—figure supplement 1C*).

Because chromosomal aneuploidy has been associated with unrepaired DNA double-strand breaks (DSBs) (*Turner et al., 2004*), we stained for the DSB marker γH2AFX (*Rogakou et al., 1998*).

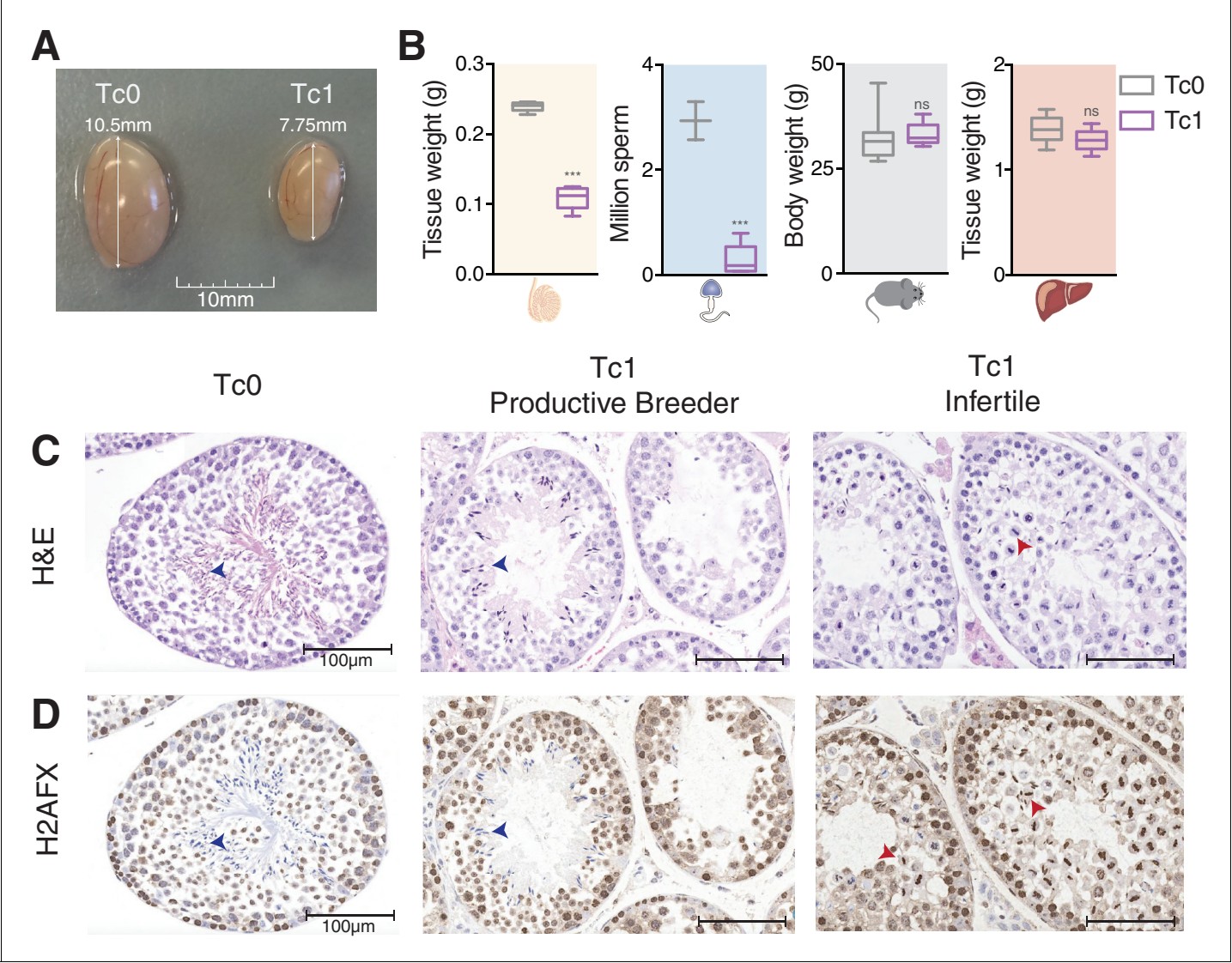

**Figure 1.** Male mice carrying human chromosome 21 show subfertility phenotypes. (A) Comparison of testis size from 12-week-old Tc0 and Tc1 littermates. (B) Testis weight, sperm count, body weight and liver weight from Tc0 and Tc1 mice. Five mice of each genotype aged between 12–14 weeks were used for tissue and body weight measurements. Sperm samples were from mice aged between 16–32 weeks, and were counted for 2 Tc0 and 5 Tc1 animals. Statistical analysis was a student's t-test (p<0.0001). Photomicrographs of testis tissue sections from adult Tc0 and Tc1 mice stained with H&E (C) and IHC with anti-γH2AFX antibody (D) original magnification 20x. Blue arrowheads indicate mature sperm, red arrowheads indicate failure in chromosome segregation. Infertile males did not produce any offspring over 6 month period kept with the same female.

The following figure supplements are available for figure 1:

**Figure supplement 1.** Tc1 mouse testes have histological abnormalities that interfere with proper spermatogenesis.

**Figure supplement 2.** γH2AFX staining of Tc1 male testes shows a higher number of double strand breaks persisting into the pachytene stage of meiosis when compared to wild-type Tc0 littermates.

In wild-type mice, we confirmed that γH2AFX is found homogeneously across the nucleus in cells undergoing meiotic recombination at the leptotene stage, followed by its restriction to the XY body in pachytene cells (*Hamer, 2002*; *Mahadevaiah et al., 2001*) (*Figure 1D* and *Figure 1—figure supplement 2A*). In contrast, Tc1 mice show a wider and more diffuse distribution of γH2AFX that does not appear to be restricted to leptotene cells (*Figure 1D* and *Figure 1—figure supplement 2B*).

The strong γH2AFX staining detected in metaphase I cells could be caused by increased apoptosis (*Odorisio et al., 1998*) indicating a potential arrest at this stage as previously suggested (*Cloutier et al., 2015*).

In sum, Tc1 males display a subfertility phenotype but produce mature spermatozoa in a substantial subset of seminiferous tubules.

## Meiosis in Tc1 males arrests at metaphase I and displays chromosome congression defects

To investigate the potential meiotic arrest at metaphase I, we staged seminiferous tubules stained using the Periodic Acid Schiff (PAS) technique, according to a binary decision code (*Meistrich and Hess, 2013*), see also Materials and methods). Due to the reduced number of spermatids in Tc1 animals, the distinction between tubules in early and middle stages of spermatogenesis was often not as clear as for wild-type animals; however, stage XII tubules were reliably identified based on their meiotic figures (*Figure 2—figure supplement 1*). This classification revealed a significant increase in the percentage of stage XII tubules in Tc1 males, supporting an arrest at metaphase I (*Figure 2A*).

To confirm these observations, we used immunofluorescence (IF) to identify phosphorylated histone H3 on Serine 10 (pH3) and α-tubulin and performed an interactive, learning-based quantification of different cell types using ilastik (*Sommer et al., 2011*) (*Figure 2B and C*). Cell identity was manually annotated to train the classifier, which was subsequently used to distinguish between cells of the germinal epithelium (purple), primary (4N) spermatocytes (green), meiotic cells (red) as well as round and elongating spermatids (light and dark blue, respectively) (*Figure 2C*, *Figure 2—figure supplement 2A and B*). Separate identification of spermatogonia or Sertoli cells was not possible using this approach, resulting in these cells being classified as either 4N spermatocytes or round spermatids depending on cell size. We expect any mis-identification to be consistent between Tc1 and wild-type mice, as neither cell type appeared to be affected by the presence of the human chromosome. In contrast, quantification revealed a significant increase in the percentage of primary spermatocytes in Tc1 animals over wild-types and a corresponding decrease in both round and elongating spermatids.

Meiotic cells have a high level of pH3 staining, a histone modification commonly found on metaphase chromosomes (*Wei et al., 1999*); on average, the percentage of meiotic cells was almost twice as high in Tc1 animals. pH3-positive cells were manually classified into meiotic stages based on DNA condensation and their spindle structure (*Figure 2D*, *Figure 2—figure supplement 2C and D*). Despite variability especially amongst Tc1 animals, we observe an almost 2-fold increase in the ratio of metaphase I to pro-metaphase cells (1.02 for Tc1 and 0.52 for Tc0), supporting an arrest at metaphase I. Interestingly, we observed abnormal metaphase cells with congression defects more frequently in Tc1 males compared to wild-types, which may contribute to the metaphase I arrest (*Figure 2E*, *Figure 2—figure supplement 2D*).

Finally, we observed a drastic increase in the percentage of cells displaying positive staining for cleaved caspase-3 within stage XII tubules of Tc1 males (*Figure 2F*), consistent with increased apoptosis due to the activation of the spindle checkpoint at metaphase I (*Eaker et al., 2001*). This increase parallels the increased staining observed for yH2AFX.

Thus, our results demonstrate that spermatogenesis in Tc1 males is impaired primarily at epithelial stage XII due to the activation of the spindle checkpoint at metaphase I.

## Efficient passage of a complete human chromosome through mouse male meiosis

We then asked whether mouse sperm containing 42 MB of the transchromosomic HsChr21 can successfully fertilize a wild-type egg and produce aneuploid offspring. Numerous trisomic mouse models and transchromosomic mouse strains have reported that the transmission of extra-chromosomal material through the male germline is difficult, if not impossible, depending on the size of the exogenous DNA (*Davisson et al., 2007*; *O'Doherty et al., 2005*; *Voet et al., 2001*). Indeed, the established protocol to passage HsChr21 is via the female germline by breeding Tc1 females with wild-type males (129S8 x C57BL/6J F1).

We first confirmed that successful transmission of human chromosome 21 occurs in 35% of offspring born to aneuploid Tc1 mothers (*O'Doherty et al., 2005*) (*Figure 3*). We produced 824

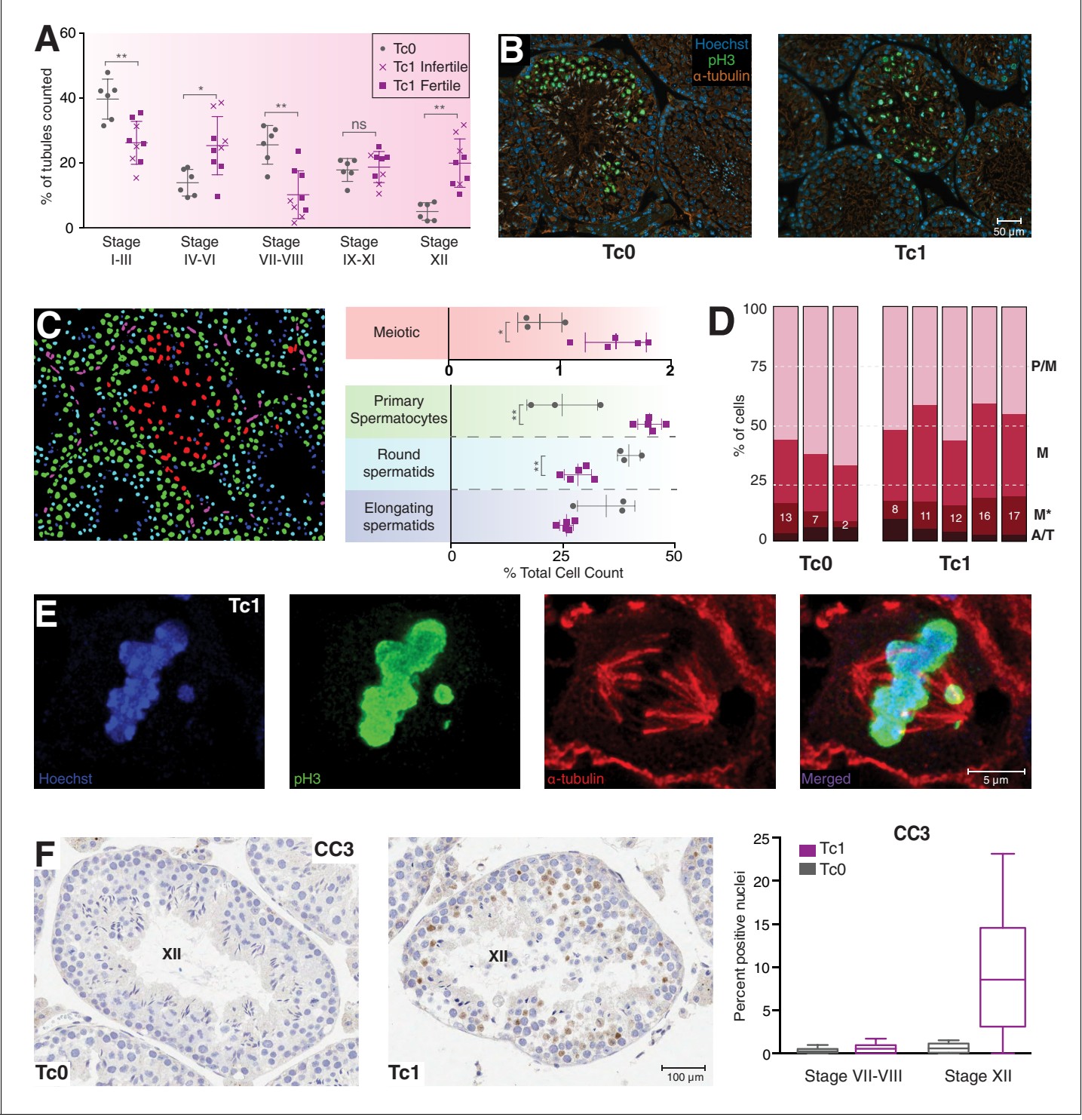

**Figure 2.** Meiosis in Tc1 males arrests at metaphase I and shows congression defects. (**A**) Percentage of tubules per spermatogenic stage are shown for individual Tc0 and Tc1 males. Infertile males did not produce offspring over a six month period kept with the same female. Statistical analysis was a student's t-test (*p<0.05; **p<0.005). (**B**) Representative immunofluorescent images of stage XII tubules from Tc0 and Tc1 animals stained with Hoechst in blue, anti-phospho histone H3 on serine 10 (pH3) in green, anti-α-tubulin in red. (**C**) An illustration using the Tc1 tubule presented in (**B**) of the annotation generated by the interactive learning algorithm which was used to quantify different cell populations from all animals to generate the % cell counts shown. Cells of the germinal epithelium (purple), 4N spermatocytes (green), meiotic cells (red) as well as round spermatids (light blue) and elongating spermatids (dark blue) were quantified for each individual animal. Statistical analysis was a student's t-test (*p<0.05; **p<0.005). (**D**) Manual quantification of the percentage of cells in different meiotic stages (pro-metaphase (P/M), metaphase (M) abnormal metaphase (M*), ana- and

*Figure 2 continued on next page*

*Figure 2 continued*

telophase (A/T) for individual animals. (**E**) Representative confocal image of a Tc1 metaphase cell with congression defect. Scale bar is 5 µm. (**F**) Representative tissue sections from Tc0 and Tc1 testes stained with anti-CC3 (cleaved caspase-3) antibody. Original magnification 20x; scale bar is 100 µm. Nuclei positive for cleaved caspase-3 were quantified in epithelial stage VII-VII and stage XII tubules.
The following figure supplements are available for figure 2:

**Figure supplement 1.** Staging of seminiferous tubules in Tc0 and Tc1 animals on periodic acid schiff (PAS)-stained tissue sections.
**Figure supplement 2.** Quantification of different cell populations and identification of meiotic cells in seminiferous tubules

offspring in 153 litters from 38 actively breeding Tc1 female mice, 290 of these offspring were Tc1 positive (*Figure 3—figure supplement 1*).

Remarkably, parallel breeding experiments with aneuploid Tc1 males, where the human chromosome is passaged via sperm, produced successful transmission of human chromosome 21 in 11% of offspring born to wild-type mothers. Twenty-seven breeding Tc1 males produced 138 litters with a total of 910 offspring, 98 of which were Tc1 positive. Male germline-derived offspring are macroscopically indistinguishable from female germline-derived offspring and no novel histological alterations were identified across different tissues (*Figure 3—figure supplement 2*). Despite the subfertility phenotypes we observed for Tc1 males, the majority of males used for breeding (21 out of 27) were fertile. 19 out of these 21 breeders transmitted human chromosome 21 at least once to their offspring, with individual breeders showing varying transmission rates and inconsistent litter frequencies (*Figure 3—figure supplement 3*). Male germline-derived offspring are themselves fertile and can successfully transmit human chromosome 21.

To see whether the comparably low rates of transmission via males originates from a specific loss of aneuploid cells during meiosis, we genotyped meiotic cells using fluorescence *in situ* hybridization (FISH). Distinct meiotic cell populations were identified and isolated based on DNA content using fluorescence activated cell sorting (FACS) (*Bastos et al., 2005*) (*Figure 3—figure supplement 4A*). The cell profiles obtained for wild-type and Tc1 males further confirmed our previous quantification, showing an increase in 4N spermatocytes and a reduction in round and elongating spermatids in Tc1 males (*Figure 3—figure supplement 4B–D*). To determine the percentage of cells carrying human chromosome 21 before and after the two consecutive meiotic divisions, we genotyped 4N spermatocytes and 1N spermatids using a probe specific for HsChr21 (*Figure 3—figure supplement 4E–G*). Almost all of the 4N spermatocytes (~94%) were positive for HsChr21, showing a surprisingly low level of mosaicism compared to previously published rates in somatic tissues (*O'Doherty et al., 2005*; *Wilson et al., 2008*). Among the haploid population, approximately 34% of round and elongating spermatids carried HsChr21. Since the best-case scenario is that 50% of haploid mouse cells carry the aneuploid human chromosome, our results suggest only a modest loss of HsChr21 during male meiosis. However, this loss cannot fully account for the low transmission rate of HsChr21 we observe in males.

Although occurring at an appreciably lower frequency than via female germline transmission, our results demonstrate conclusively that the mouse male germline can successfully package an exogenous and aneuploid 42 MB human chromosome into protamines to generate reproductively active sperm.

## Accurate and precise transcription initiation in adult tissues of a human chromosome that has been passaged through mouse spermatogenesis

We asked whether the male-germline specific process of stripping the human chromosome of the vast majority of its histones, followed by its reconstruction post-fertilization using mouse epigenetic machinery, impacted the transcriptional deployment of the chromosome in derived adult mouse tissues.

We first compared sites of transcriptional initiation across female and male germline-derived human chromosome 21. As a proxy for transcriptional activation, we mapped trimethylation of lysine 4 on histone 3 (H3K4me3) (*Bernstein et al., 2006*; *Guenther et al., 2007*; *Heintzman et al., 2007*)

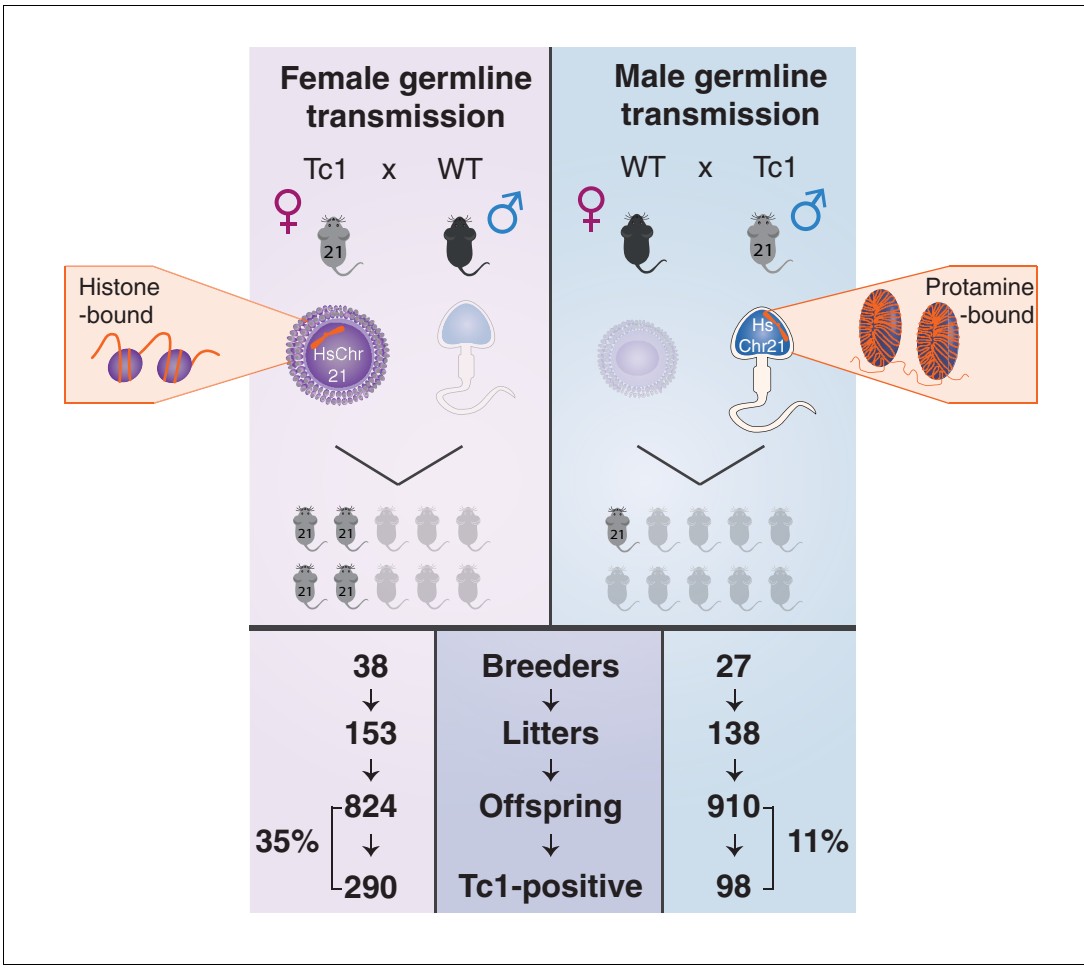

**Figure 3.** Male germline transmission of human chromosome 21 is a third as efficient as female germline transmission. Oogenesis (left, top panel) can allow the transmission of epigenetic information deposited on retained maternal histones, whereas the majority of histones are replaced by protamines during spermatogenesis (right, top panel). Germline transmission of the full aneuploid chromosome HsChr21 was successful using male Tc1 mice as transmitters, but at a substantially reduced frequency compared with female transmission via eggs (11% versus 35%).

The following figure supplements are available for figure 3:

**Figure supplement 1.** Transmission rates of aneuploid human chromosome 21 when passaged by the eggs of breeding Tc1-positive females.

**Figure supplement 2.** Histology across multiple somatic tissues in wild-type mice and female and male germline-derived Tc1 mice.

**Figure supplement 3.** Transmission rates of aneuploid human chromosome 21 when passaged by the sperm of breeding Tc1-positive males.

**Figure supplement 4.** Genotyping of meiotic cells from Tc1 males showed a high percentage of aneuploid haploid cells.

using chromatin immuno-precipitation followed by high-throughput sequencing (ChIP-Seq) in a number of adult somatic tissues.

Across the entire human chromosome, we observed neither qualitative nor quantitative differences in the locations of transcription initiation in endoderm-derived livers from male and female

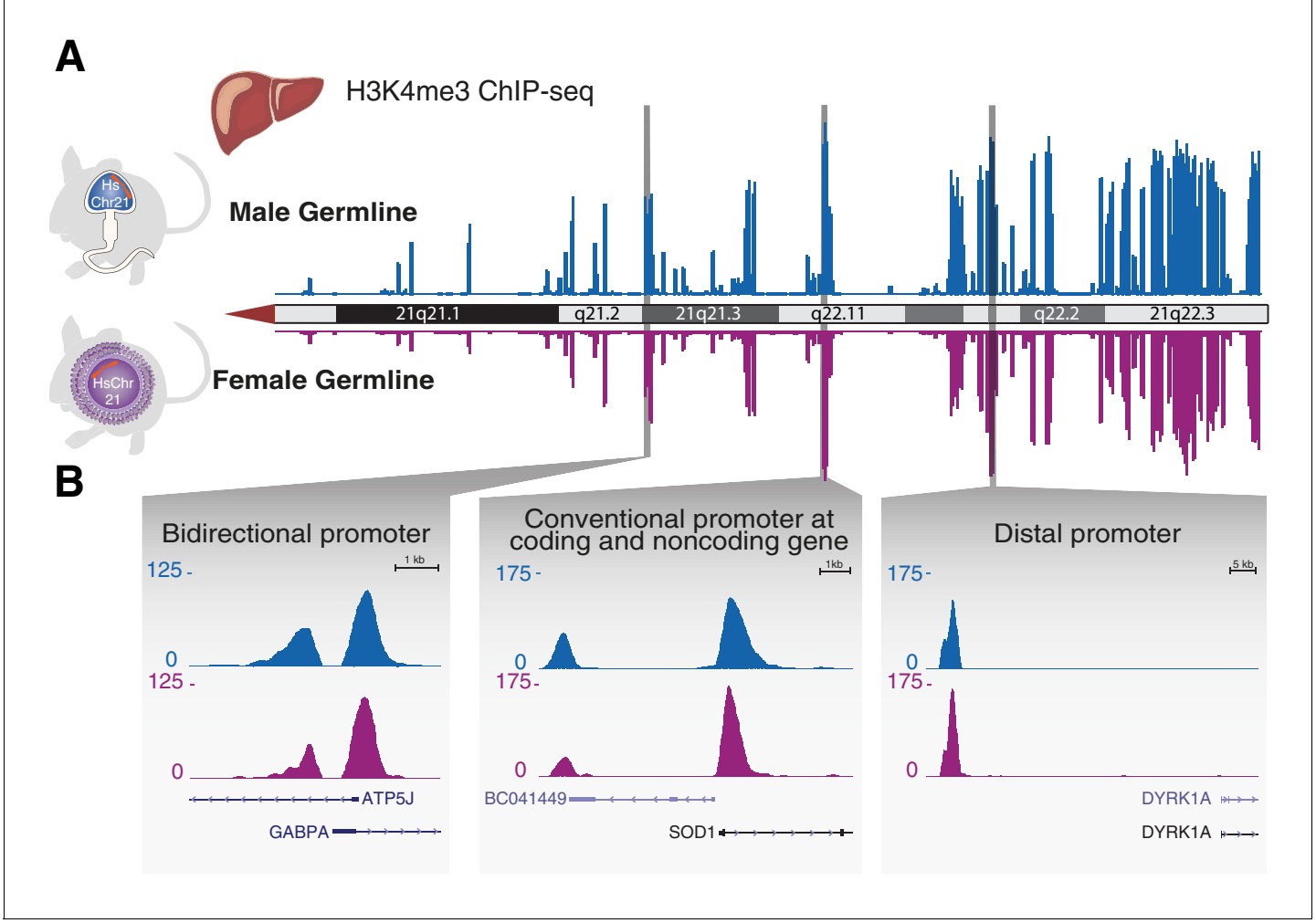

**Figure 4.** Transcription initiation across aneuploid human chromosome 21 in terminally differentiated liver is unaffected by the differing epigenetic handling during spermatogenesis or oogenesis. (A) The occupancy of H3K4me3, which reports transcription initiation locations, is shown as an enrichment track across the q arm of human chromosome 21 from livers of Tc1 mice derived from aneuploid sperm (blue) or eggs (purple). (B) Transcription initiation patterns at diverse promoter types on human chromosome 21 are indistinguishable between male and female germline transmission.

The following figure supplement is available for figure 4:

**Figure supplement 1.** Transcription initiation is accurately established across multiple somatic tissues after male germline transmission.

germline-derived Tc1 mice (*Figure 4A*). The stability of the eventual intensity of transcription initiation was found at all unidirectional and bidirectional promoters, regardless of their proximal or distal positioning relative to coding and noncoding genes (*Figure 4B*).

Finally, we mapped the sites of transcription initiation in multiple tissues representing the other two germ layers, including kidney (mesoderm), muscle (mesoderm) and brain (ectoderm). In all profiled tissues, the large-scale remodelling caused by male germline passage of the human chromosome resulted in transcription initiation indistinguishable from female germline-derived HsChr21 (*Figure 4—figure supplement 1*). These results strongly argue that the process of chromatin decondensation (and complete rebuild by the mouse machinery) after fertilization does not distort the transcriptional deployment of the aneuploid human chromosome during mouse embryogenesis.

## Human repeats on the male germline passaged human chromosome

The process of spermatogenesis can transiently unmask repetitive elements on a transcriptional level (*Carmell et al., 2007*). We therefore asked whether passage of the HsChr21 through the Tc1 male spermatogenic programme could cause persistent activation of human-specific repeat elements through fertilization and development.

Our previous comparisons between Tc1 mice and humans in liver showed that the mouse nuclear environment unmasks a number of human- and primate-specific repetitive elements across chromosome 21 passaged through the female germline (*Ward et al., 2013*). Because spermatogenesis and later post-fertilization depackaging involves substantial transient transcriptional activation of repetitive elements (*Carmell et al., 2007*; *Fadloun et al., 2013*), we asked whether human repeats on HsChr21 transmitted via sperm would retain enhanced activation in derived Tc1 somatic tissues. However, the previously-reported mouse-specific transcription initiation events located at human repeat elements were accurately regenerated when the aneuploid human chromosome was passed through the male germline (*Figure 5*).

## Massive male germline-specific epigenomic remodelling results in stable transcriptional deployment of the human chromosome

We considered the possibility that the process of decondensation could adversely affect other layers of transcriptional control on the human chromosome, including enhancers, promoters, DNA methylation, tissue-specific transcription factor (TF) binding, and the core transcriptional machinery. Using liver as a representative somatic tissue, in addition to H3K4me3 (marking active transcription initiation at promoters) we epigenetically profiled the genomic occupancy of H3K27ac (active enhancer regions), two tissue specific TFs (CEBPA (CCAAT/enhancer-binding protein alpha) and HNF4A (hepatocyte nuclear factor 4 alpha)) and RNA polymerase 2 (Pol II) (the basal transcriptional machinery), as well as total RNA-Seq (active transcription) and non-methylated DNA across human chromosome 21 in female and male germline-derived Tc1 mice (*Figure 6*).

Promoter and enhancer activity is highly correlated between male and female germline-derived mice ($r^2$ = 0.99 and 0.96, respectively) with no significantly differentially bound sites on chromosome 21 (*Figure 6A,B*). We used the presence of the entire mouse genome in both Tc1 samples as internal technical controls to evaluate what genome-wide correlation in transcription initiation would be expected between diploid individuals. Differential binding analysis across the mouse genome revealed a total of 10 out of 20,001 sites for H3K4me3 and 471 out of 47,254 sites for H3K27ac that showed a fold change greater than 2.5 (FDR < 0.1) (*Figure 6—figure supplement 1A,B*). These modest differences did not appear to be due to the presence of HsChr21, as these numbers are comparable to technical noise levels we calculated using previously published wild-type mouse replicates for H3K27ac (599 out of 41,327) (*Figure 6—figure supplement 1C*) (*Villar et al., 2015*).

To identify whether different germline passaging would lead to alterations in the DNA methylation underlying the chromatin, we identified hypomethylated regions on human chromosome 21 passaged via either egg or sperm using BioCAP-sequencing (biotinylated CxxC affinity purification) (*Blackledge et al., 2012*; *Long et al., 2016*). As with the chromatin marks above, DNA methylation in adult somatic tissues converges on the same molecular phenotype ($r^2$ = 0.97) (*Figure 6C*).

Unsurprisingly, the correlation in genomic occupancy between male and female germline-derived mice for two liver-specific transcription factors (CEBPA and HNF4A) and Pol II were slightly noisier ($r^2$ = 0.72–0.83) (*Stefflova et al., 2013*); nevertheless, almost no sites were identified reliably as differentially bound (*Figure 6D-F*). In most cases, differences were due to modest changes in overall ChIP intensity, not the occurrence of entirely novel occupancy in male- or female-derived Tc1 mice. The noisier correlation of Pol II is likely due to the more distributed nature of polymerase genome occupancy: Pol II typically binds across tens of kilobases at comparatively low intensity, as opposed to more sharply defined regions occupied by modified histones in active regions of the genome. Notably, the modest differences observed in polymerase occupancy do not impact the transcriptome, which shows exceptionally high correlation between livers of Tc1 female- and male-derived offspring ($r^2$ = 0.97), with no genes identified as differentially expressed (*Figure 6G*).

Overall, despite the massive epigenetic remodelling and chromatin condensation associated with male germline transmission, human chromosome 21 is accurately deployed in multiple diverse tissues during development by the mouse epigenetic machinery.

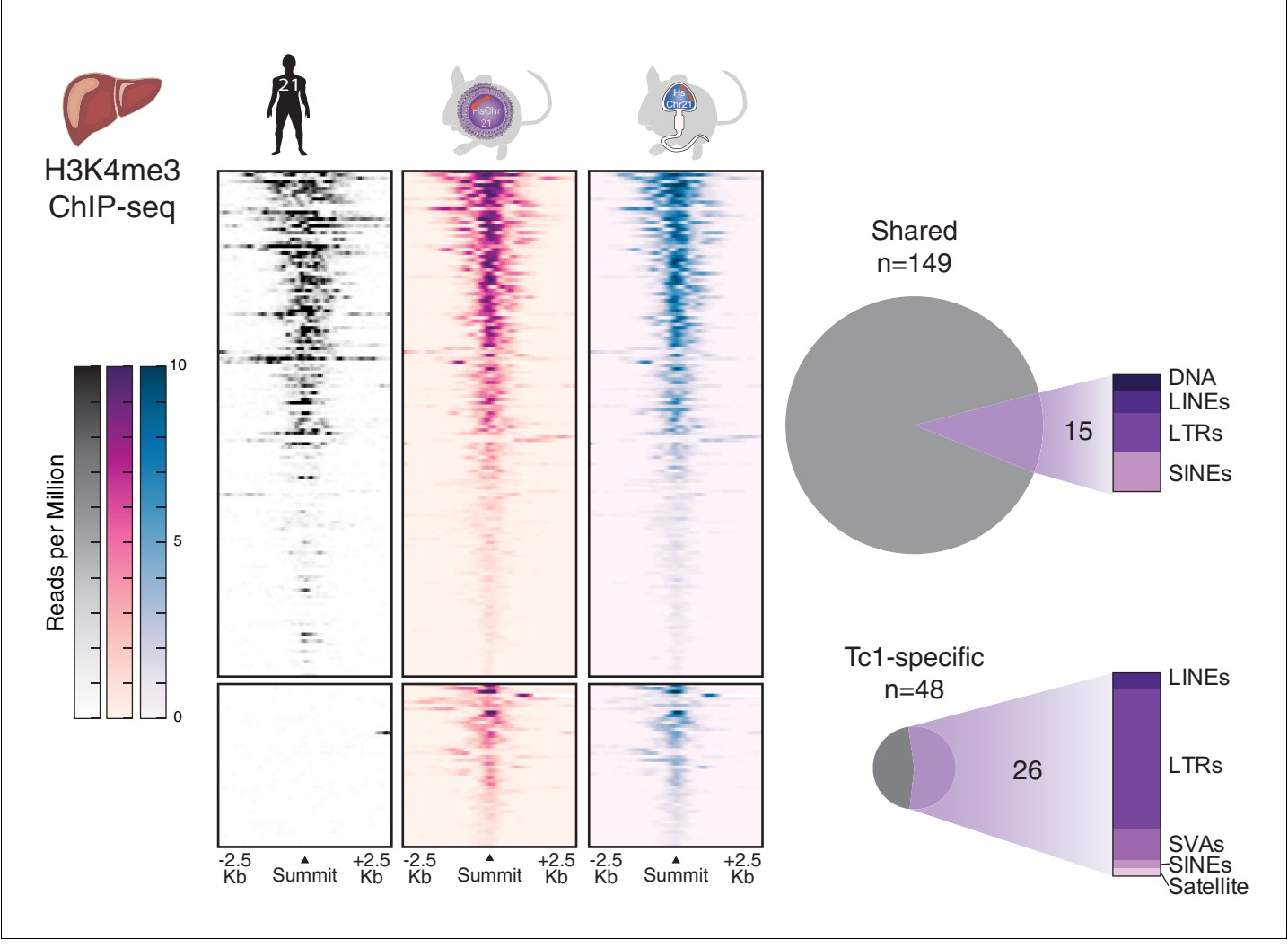

**Figure 5.** Comparison of transcription initiation between human and Tc1 mouse reveals that male germline passage does not unmask novel human repetitive elements in derived mouse somatic tissues. Differential binding analysis of transcription initiation locations between human and Tc1 mouse liver reveals Tc1 mouse-specific sites, which were indistinguishable by germline passage. No entirely novel male-germline sites were identified. Regions in heatmaps are sorted by descending signal strength, each row representing a 5 kb window centered on H3K4me3 peak summits. More than half of the mouse-specific sites are found at human repetitive elements.

## Discussion

The creation of haploid gametes is one of the most tightly regulated processes in cell biology, as failure to accurately evaluate DNA content can result in catastrophic organismal aneuploidies. Embryos containing aneuploidies are spontaneously aborted during development, though a few, such as in humans with Down syndrome, can be tolerated despite developmental defects (*Hassold and Hunt, 2001*).

Aneuploid female mice that carry large amounts of exogenous DNA are fertile and can pass aneuploid DNA on to their offspring; however, aneuploid male mice have strongly suppressed fertility and are often entirely sterile (*Co et al., 2000*; *Hernandez and Fisher, 1999*). Male sterility is commonly observed amongst transchromosomic mouse models, and is attributed to the presence of an extra chromosome rather than the trisomic gene content, causing spermatogenic arrest at metaphase I (*Hernandez and Fisher, 1999*).

How Tc1 male testes handle human chromosome 21 during spermatogenesis, particularly during prophase I, has been previously studied (*Mahadevaiah et al., 2008*). In more than 50% of the cases,

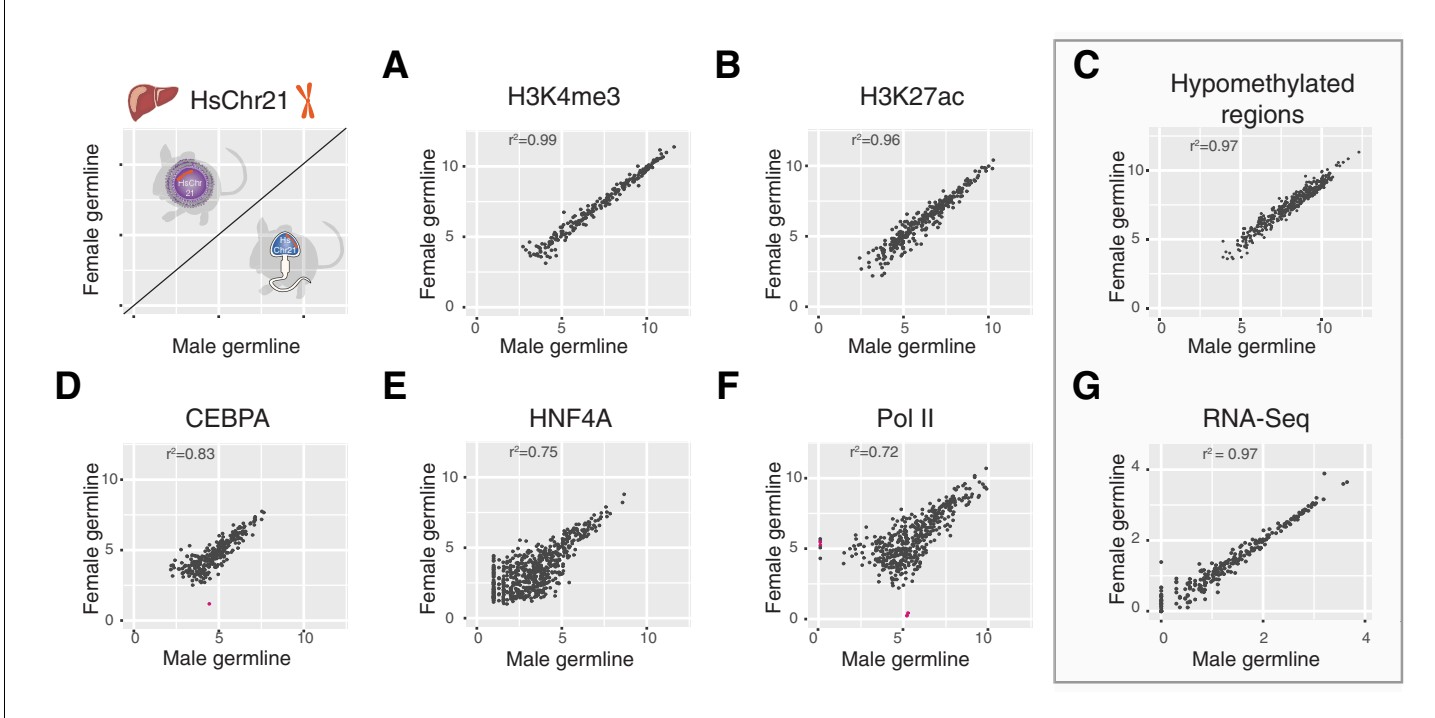

**Figure 6.** Multiple layers of transcription and transcriptional control across human chromosome 21 are indistinguishable between Tc1 mice whether derived from aneuploid eggs or sperm. Comparison of ChIP-Seq mean read concentration (log2) across human chromosome 21 in livers of female and male germline transmitted offspring for (**A**) H3K4me3 (transcription initiation), (**B**) H3K27ac (enhancer activity), (**D**) CEBPA (tissue-specific transcription factor), (**E**) HNF4A (tissue-specific transcription factor) and (**F**) RNA polymerase II. Differentially bound sites are highlighted in pink (fold change >2.5, FDR <0.1). Pearson correlation was applied to analyse the correlation. (**C**) Comparison of genome hypomethylation patterns between male and female germline offspring profiled by BioCAP-sequencing (log2 read count). (**G**) Differential gene expression analysis of RNA-Seq in liver between male and female germline-derived offspring (log10 mean expression).

The following figure supplement is available for figure 6:

**Figure supplement 1.** Differential binding analysis for H3K4me3 and H3K27ac across the mouse genome in Tc1 and BL6 mice.

human chromosome 21 is incorporated into the large γH2AFX domain together with X and Y chromosome to mask their incomplete synapsis, resulting in transcriptional silencing (*Handel, 2004*). However, independent of whether human chromosome 21 would be unsynapsed and in proximity to the XY body or self-synapsed and located away from XY, the presence of human chromosome 21 did not lead to increased pachytene apoptosis in epithelial stage IV tubules (*Mahadevaiah et al., 2008*). Whether this would allow the production of viable sperm and the transmission of chromosome 21 to the next generation, however, was not tested.

The first successful passage of human DNA via mouse germline followed the development of microcell-mediated chromosome transfer of human chromosome fragments into mouse ES cells. Mice derived from these ES cells were able to passage fragmented regions of human DNA via the female and occasionally male germline (*Tomizuka et al., 1997*). To date, the largest successful and stable germline transmission of human DNA via mouse sperm was of a circularized 5–10 megabase human artificial chromosome (*Voet et al., 2001*); a linearized version of this same artificial chromosome containing short telomeres can also be passaged via sperm (*Weuts et al., 2012*).

We demonstrated that mouse male meiosis can produce viable sperm that both carries and transmits the complete 42 MB copy of human chromosome 21 to generate viable aneuploid offspring. Consistent with the reduced fertility described above for aneuploid males, Tc1 testes show macro- and microscopically visible disruptions to their tissue architecture. Nevertheless, despite frequent spermatogenic arrest at metaphase I leading to increased apoptosis, the majority of Tc1 males tested in this study were able to produce viable aneuploid offspring, albeit at lower frequencies. The

observed lower transmission rate of HsChr21, however, cannot be fully accounted for by the reduced number of HsChr21 positive cells produced during male meiosis. This indicates that failures are likely to occur further downstream, i.e. during sperm maturation, at fertilization or during early embryonic development, eventually resulting in only ~ 11% aneuploid offspring. If the deficit in Tc1-positive offspring arises post-fertilization, then these embryos lost during early development may have been unable to accurately deploy the Tc1 chromosome. While female meiosis is more error-prone than male meiosis due to weaker checkpoint mechanisms (*Morelli and Cohen, 2005*), the percentage of aneuploid cells produced via male meiosis (34%) is strikingly similar to the percentage of aneuploid offspring generated via female germline transmission.

Attempts to establish the Tc1 chromosome on three different inbred genetic backgrounds led to a complete loss of HsChr21 over only a few generations (*O'Doherty et al., 2005*). The loss of the human chromosome was independent of the sex of the transmitting parent, and stable transmission was only observed when Tc1 mice were crossed with F1 hybrids between BL6 and 129S8 mice. It is currently unclear why a heterozygous genetic background is necessary to maintain stable transmission of HsChr21 or to what extent strain-specific checkpoint mechanisms may be involved. Studies in XO females have suggested that C3H mice have a weaker spindle assembly checkpoint (SAC) than C57/BL6 (*LeMaire-Adkins and Hunt, 2000*; *Nagaoka et al., 2011*); no equivalent studies are available for the 129S8 strain. However, few transcriptional differences exist between the testes of 129S8xBL6 F1 hybrids (Tc0) and inbred wild-type BL6 mice (data not shown).

The Tc1 mouse has long been an elegant model of human Down syndrome that recapitulates many clinical features of human trisomy 21 (*O'Doherty et al., 2005*). Histologically, the subfertility phenotypes we observed for the Tc1 mouse may be similar to developmental abnormalities reported for human males trisomic for chromosome 21 (*Johannisson et al., 1983*). Down syndrome, especially when paternally inherited, exhibits a strong sex bias (up to a 3.5 male/female ratio), which has been attributed to a preference of the extra chromosome to segregate with the Y chromosome (*Hassold et al., 1984*; *Nicolaidis and Petersen, 1998*; *Petersen et al., 1993*). In contrast, however, we did not observe a sex bias amongst aneuploid offspring from either female or male germline transmission (1.0 and 1.1, respectively), possibly reflecting interspecies differences in segregation patterns.

The successful passage of the human chromosome through the male germline held the potential to further unmask human-specific repetitive elements, because of the genome-wide transcriptional activation occurring during spermatogenesis and the (assumed) absence of species-specific mechanisms that co-evolved to repress human repeats (*Jacobs et al., 2014*; *Zamudio and Bourc'his, 2010*). However, no additional repeat elements appeared to be transcriptionally activated in somatic tissues following male germline transmission. Instead, the massive epigenomic remodelling associated with histone removal during spermatogenesis and the subsequent rebuilding post fertilization resulted in the same patterns of DNA hypomethylation, transcriptional activation, enhancer deployment, transcription factor binding, and RNA transcription in male-derived offspring as was found in female germline passaged mice.

Our results reveal the remarkable insight that mouse male meiosis can tolerate the presence of an aneuploid 42 MB human chromosome to generate viable sperm. The radically differing developmental dynamics of male- versus female-germline passage of this aneuploid chromosome can nevertheless result in indistinguishable transcriptional and regulatory phenotypes.

## Materials and methods

### Mouse material

The Tc1 mouse line was obtained from Dr. E. Fisher and Dr. V. Tybulewicz (*O'Doherty et al., 2005*) and housed in the Biological Resources Unit (BRU) in the Cancer Research UK – Cambridge Institute under the Home Office Licence (PPL 70/7535). For maintenance of the transgenic line, the human chromosome 21 (HsChr21) was transmitted through the female germline by breeding female Tc1 mice to male (129S8 x C57BL/6J) F1 mice (conventional breeding setup). For male germline transmission, female (129S8 x C57BL/6J) F1 mice were crossed with male Tc1 mice. Tc1-negative littermates (Tc0) were used as control animals. Tissues were obtained from at least two independent males at an age between 8–12 weeks and were either flash frozen for RNA-Seq and BioCAP-Seq,

cross-linked with 4% formalin for histology or cross-linked with 1% formaldehyde for ChIP-Seq as previously described (*Schmidt et al., 2009*).

Mouse sperm was obtained from male mice at an age between 16–32 weeks according to *Hisano et al. (2013)* and quantified using a hemocytometer.

## Histology

Tissues were fixed in neutral buffered formalin (NBF) for 24 hr, transferred to 70% ethanol, machine processed and paraffin embedded. All formalin-fixed paraffin-embedded (FFPE) sections were 3 μm in thickness and stained with haematoxylin and eosin (H & E) for morphological assessment and classification. Slides were scanned using Aperio XT (Leica Biosystems, UK).

Immunohistochemistry (IHC) was performed on FFPE sections using the Bond™ Polymer Refine Kit (DS9800, Leica Microsystems) on the automated Bond platform. De-waxing and re-hydration prior to IHC and post-IHC de-hydration and clearing were performed on the automated Leica ST5020; mounting was performed on the Leica CV5030. Antibodies against phospho-Histone H2A.X (Ser139) (Millipore, MABE205, 1:5000 dilution) and cleaved Caspase-3 (Cell Signaling Technology, 9664, 1:200 dilution) were used with DAB Enhancer (Leica Microsystems, AR9432). Heat-induced epitope retrieval was performed for 20 min at 100°C on the Bond platform with sodium citrate (for γH2AFX) and Tris EDTA (for CC3). Slides were scanned using Aperio XT (Leica Biosystems) and CC3 quantification was performed using the Aperio eSlide Manager (Leica Biosystems).

## Immunofluorescence staining of seminiferous tubules

FFPE sections (standard 3 μm sections for widefield microscopy and thicker 10 μm sections for confocal microscopy) were de-waxed on the automated Leica ST5020 and antigen retrieval was performed by boiling the slides for 10 min in 10 mM sodium citrate + 0.05% Tween-20. Sections were permeabilized in PBS + 0.3% Triton-X100 for 30 min and blocked with 5% BSA in PBS + 0.3% Triton-X100 for 1 hr. Incubation with primary antibodies against α-tubulin (Sigma, USA, T9026, 1:500 dilution) and phospho-Histone H3 (Ser10) (Millipore, 06–570, 1:1000 dilution) in 5% BSA in PBS + 0.1% Tween-20 was performed for 2–3 hr at 37°C in a humidifying chamber. Slides were washed in PBS + 0.1% Tween-20 and incubated with secondary antibodies anti-mouse IgG conjugated to AlexaFluor-488 and anti-rabbit IgG conjugated to AlexaFluor-555 (ThermoFisher Scientific, UK, A21206 and A31570) for 1 hr at room temperature. Slides were washed, stained with Hoechst 33342 (1 μg/ml) for 15 min and mounted in ProLong Diamond Antifade Mountant (ThermoFisher Scientific).

Widefield microscopy of tissue sections was performed using a Zeiss Axio Obsever Z1 with a Pl APO 0.8 NA 20X dry objective (Carl Zeiss Microscopy, DE) fitted with a CoolLED PE-4000 LED lightsource and Zeiss Axiocam 506 camera. A 2D tile-scan across the entire tissue section was performed with 10% tile overlap. Voxel size was 0.23 μm.

Confocal microscopy was performed using a Leica TCS SP8 STED 3X microscope with an HC PL APO CS2 1.4NA 100X oil objective (Leica Microsystems). A 405 nm diode laser was used to excite Hoechst at 405 nm and a white light pulsed laser (SuperK EXTREME, NKT Photonics, DK) was used to excite the secondary antibody fluorophores. Voxel size was 0.07 μm and a Z-stack was acquired through the sample with 0.3 μm spacing. Each channel was acquired sequentially. Post-acquisition the data deconvolved using Huygens Professional (Scientific Volume Imaging, Version 15.10.1p2).

## Interactive learning using ilastik

The open source interactive learning toolkit ilastik was used to segment and classify cells in the stained tissue sections (*Sommer et al., 2011*). This was done with a two-stage process: pixel classification followed by object classification. Pixel classification was performed to segment the nuclear regions, and object classification to split each nucleus into one of five classes: germinal epithelium, primary spermatocytes, meiotic, round spermatids and elongating spermatids. The training data contained two cropped images from each tissue section and image annotation was done blindly with respect to condition. Each tile was classified independently and Matlab (2015b, MathWorks) was used to process the results and remove duplicated objects.

The subset of cells identified as meiotic by the ilastik toolbox were manually classified as either pro-metaphase, metaphase, phenotypic metaphase, anaphase or non-mitotic. A randomly selected subset of 200 meiotic cells were analysed from each section. To eliminate user bias and facilitate

blind analysis, the order of the cells was randomised. This was performed using Matlab (2015b, MathWorks) with a bridge to ImageJ for visualisation (*Sage et al., 2012*).

## Purification of meiotic cell populations using fluorescence-activated cell sorting

Spermatogenic cells were isolated from adult mouse testes (16–36 weeks old) as described in *Goh et al. (2015)* with minor modifications. One testis from wild-type males and both testes from Tc1 males were used per experiment. In brief, the albuginea was removed and tissue was digested in dissociation buffer (25 mg/ml Collagenase A, 25 mg/ml Dispase II and 2.5 mg/ml DNase I) for 30 min at 37°C. Enzymatic digestions were quenched with DMEM + 10% FCS, resuspending the cells at a concentration of 1 million cells per ml. Hoechst 33342 was added to a final concentration of 5 µg/ml and stained for 45 min at 37°C in the dark. Cells were resuspended in PBS + 1% FCS+2 mM EDTA for sorting and propidium iodide was added to a final concentration of 1 µg/ml for dead cell exclusion. Cells were sorted on an Aria IIu cell sorter (Becton Dickinson, UK) using a 100 µm nozzle. Hoechst was excited with a UV laser at 355 nm and fluorescence was recorded with a 424/44 filter (Hoechst blue) and 675LP filter (Hoechst red). Four different cell populations distinguished by DNA content were sorted corresponding to cells in S phase, primary spermatocytes (4N), secondary spermatocytes (2N) and spermatids (1N) were collected into PBS + 1% FCS+2 mM EDTA.

## Fluorescence *in situ* hybridization of human chromosome 21 in meiotic cells

Meiotic cell populations obtained from FACS sorting were spun onto Superfrost Plus slides using a Cytospin at 1000 rpm for 3 min and fixed with methanol:acetic acid (3:1) for 30 min. Genotyping was performed by fluorescence *in situ* hybridization (FISH) using a human chromosome 21-specific probe XA 21q22 (Metasystems, DE, D-5601–100-OR). In brief, slides were treated with 0.01M HCl + 0.5 ug/ml pepsin at 37°C for 10 min, washed in water and dehydrated in 70% and 10% ethanol. The probe was applied, sealed with a coverslip and denatured at 80°C for 2 min followed by hybridisation for 16 hr at 37°C. The coverslip was removed in 2X SSC + 0.05% Tween-20, slides were washed in 0.4X SSC for 2 min at 72°C, rinsed again in 2X SSC + 0.05% Tween-20 and mounted in Prolong Gold + DAPI.

Widefield microscopy was performed as described for immunofluorescence. The percentage of cells containing human chromosome 21 was quantified using FIJI (*Schindelin et al., 2012*). Representative high-resolution images were captured using a Nikon TE-2000 inverted microscope with NIS-elements software using a Plan Apochromat x100 objective and Andor Neo 5.5 sCMOS camera.

## Classification of seminiferous tubules

To assess the overall subfertility phenotype, H & E stained tissue sections of testis were scored by two blinded independent individuals (C.E. and S.J.A) using Aperio eSlide Manager (Leica Biosystems). Seminiferous tubules were scored according to the predominant histological pattern of spermatogenesis in each individual tubule. Grade I: normal spermatogenesis; Grade II: mild hypospermatogenesis (all germ cell stages present but visible meiotic disruption and suboptimal frequency of spermatozoa); Grade III: severe hypo-spermatogenesis (all germ cell stages present including occasional spermatozoa); Grade IV: maturation arrest (incomplete spermatogenesis, not beyond the spermatocyte stage) (*Creasy et al., 2012*; *Dohle et al., 2012*).

Classification of seminiferous tubules according to epithelial stage of spermatogenesis was performed on PAS stained tissue sections. Different stages were identified as described in the binary decision key by *Meistrich and Hess (2013)*. Stage I-III: Two generations of spermatids but no acrosome cap over the nucleus of round spermatids; Stage IV-VI: Two generations of spermatids and acrosomic system forming a cap over the nucleus of round spermatids; Stage VII-VIII: Two generations of spermatids with elongated spermatids lining the lumen; Stage IX-XI: Only one generation of spermatids but no visible meiotic figures or secondary spermatocytes; Stage XII: Only one generation of spermatids and visible meiotic figures as well as secondary spermatocytes.

## Chromatin immuno-precipitation followed by high-throughput sequencing (ChIP-Seq)

Mouse tissue samples were harvested after direct liver perfusion with PBS, cross-linked in 1% formaldehyde solution (v/v) for 20 min, followed by quenching of formaldehyde by addition of 250 mM glycine for 10 min. Cross-linked tissues were washed twice in PBS and either used directly for tissue lysis or frozen for storage at −80°C. Tissues were homogenized in a dounce tissue grinder, washed twice with PBS and lysed according to published protocols (*Schmidt et al., 2009*). Sonication was performed on a Misonix sonicator 3000 with a 418 tip to fragment chromatin to an average length of 300 bp. The following antibodies were used for immuno-precipitation: H3K4me3 (Millipore 05–1339 CMA304, Lot numbers 236661 and 2504863), H3K27ac (abcam, UK, ab4729, Lot numbers GR150367 and GR200563), CEBPA (Santa Cruz, USA, sc-9314, Lot L1113), HNF4A (ARP31946, Lot numbers QC22894 and QC1455(R1)100317), RNA polymerase II (abcam, ab5408, Lot number GR106949) according to Schmidt *et al.*. Immuno-precipitated DNA or 50 ng of input DNA was used for library preparation following the standard Illumina TruSeq ChIP Sample preparation protocol or the ThruPLEX DNA-Seq library preparation protocol (Rubicon Genomics, UK). Libraries were sequenced on HiSeq2000 or HiSeq2500 according to manufacturer's instructions using single-end 50 bp reads. Individual library preparations are detailed under ArrayExpress submission.

## RNA-Seq and differential expression analysis

Total RNA was extracted from flash-frozen liver using QIAzol Lysis Reagent (Qiagen, USA), DNase treated using the Turbo DNA-*free* kit (Thermo Fisher, AM1907) and depleted of ribosomal RNA (Illumina, UK, RiboZero, Epicenter) according to manufacturers instructions. Strand-specific libraries were prepared using dUTPs (*Kutter et al., 2012*) together with the Illumina TruSeq RNA Kit. RNA-Seq libraries were sequenced as 50 bp single-end reads on an Illumina HiSeq 2000.

RNA-Seq libraries were aligned against the reference genome using the Genomic Short-read Nucleotide Alignment Program (GSNAP) (*Wu and Nacu, 2010*). Tc1 libraries were aligned against GRCm38/mm10 with the addition of human chromosome 21 (hg19). Differential expression analysis was performed using edgeR (*Robinson et al., 2010*) with six biological replicates for female germline-derived animals and two biological replicates for male germline-derived animals.

## BioCAP and differential methylation analysis

BioCAP-sequencing was performed as previously described (*Blackledge et al., 2012*) in flash frozen liver samples from Tc1 mice derived of female and male germline transmission (two biological replicates each). BioCAP-Seq libraries were aligned against a composite genome containing all mouse chromosomes and human chromosome 21 (mm9 + hg19 HsChr21) using bowtie (*Langmead et al., 2009*). Hypomethylated regions of DNA (HMRs) were identified using MACS1.4 (*Zhang et al., 2008*) with settings –tsize = 50 –bw = 300 –mfold = 10,30 pvalue=1e-5 –verbose = 10 –g 4.8e + 8 against an input control. Only HMRs that were identified in both biological replicates were retained and HMRs overlapping known breakpoints or deletions of HsChr21 in the Tc1 mouse were removed.

For differential methylation analysis, HMRs obtained for female and male germline-derived Tc1 mice were merged and read counts over genomic intervals were obtained using bedtools genome-cov (*Quinlan and Hall, 2010*).

## ChIP-Seq peak calling

ChIP-Seq libraries were aligned against the reference genome using Burrows-Wheeler Aligner (BWA) (*Li and Durbin, 2009*). Human and wild-type mouse libraries were aligned against GRCh37/hg19 and GRCm38/mm10, respectively. Tc1 libraries were aligned against GRCm38/mm10 with the addition of human chromosome 21 (hg19). Regions with a mapping quality score of 0 were removed and only uniquely mapping reads were used for downstream analysis. ChIP-Seq libraries were filtered against ENCODE blacklisted regions (hg19/GRCh37 and mm9 liftover to mm10) (*Dunham et al., 2012*). Regions on human chromosome 21 that are deleted or duplicated in the Tc1 mouse were removed from both Tc1 and human libraries (*Gribble et al., 2013*). ChIP-Seq peaks were called using the Model-based Analysis of ChIP-Seq (MACS) algorithm version 2.0 (MACS2) (*Feng et al., 2012*). Concatenated input samples of higher complexity were used as control. The 'callpeak' function was specified as well as 'SPMR' to generate signal per million reads pileup files for visualization.

For broad spanning factors such as H3K27ac and RNA Polymerase II '–broad' was specified. 'macs2 bdgcmp –m FE' was used to generate signal tracks showing the fold enrichment of treatment over control. These tracks were used for visualization on the UCSC Genome Browser (*Kent et al., 2002*).

## Differential binding analysis

Differential binding analysis on enriched regions was performed as in Ross-Innes *et al.* (*Ross-Innes et al., 2012*), using the R/Bioconductor package DiffBind (version 1.14.5). Using dba.count, peaks were required to be present in at least one fourth of all replicates and reads were normalised using the Trimmed Mean of M-values (TMM) method using the effective library size after subtracting control reads (*Robinson and Oshlack, 2010*). Differentially bound sites were defined to have at least 2.5-fold difference in binding intensity with an FDR of less than 0.1 between conditions. The log2 mean read concentration as defined by DiffBind was plotted using ggplot2 (*Wickham, 2009*). The correlation between samples was calculated using Pearson's correlation.

Functional annotation for genomic regions were obtained using the R/Bioconductor package compEpiTools function GRannotateSimple (*Kishore et al., 2015*).

## Generation of ChIP-Seq intensity heatmaps

Pileup bedGraph files normalised to reads per million as generated by macs2 were used to plot ChIP-Seq intensity heatmaps. Bigwig files for ChIP and input libraries were uploaded onto Galaxy (*Afgan et al., 2016*) and input reads were subtracted from ChIP reads. Heatmaps were generated using the 'computeMatrix' and 'plotHeatmap' function from deepTools (*Ramírez et al., 2014*). Regions were sorted by decreasing signal strength in tissues from female germline-derived Tc1 mice, each row representing a 5 kb window around a H3K4me3 peak summit.

## Repeat overlap

Peak summits for shared and Tc1-specific sites were redefined in DiffBind using dba.count (summits = 25) and obtained using dba.peakset. The obtained 50 bp windows centred on H3K4me3 peak summits were then overlapped with repetitive elements on HsChr21 obtained from RepeatMasker (Smit, AFA., Hubley, R., Green P., RepeatMasker Open-3.0.1996–2010) with simple, telomeric and centromeric repeats removed.

## Acknowledgements

We thank the CRUK-CI Genomics, Bioinformatics, Flow Cytometry (M Strzelecki, R Grenfell), Light Microscopy and Histopathology (J Miller, J Jones) cores and the Biological Resources Unit for technical assistance. We thank P Tatrai and I Falciatori for helpful suggestions and technical assistance. This research was supported by Cancer Research UK (CE, CK, FC, TFR, ML, DTO), the European Molecular Biology Laboratory (NE), the Wellcome Trust (106563/Z/14/A: SJA and 098024/Z/11/Z: RJK) and the European Research Council (DTO).

## Additional information

### Competing interests

DTO: Reviewing editor, *eLife*. The other authors declare that no competing interests exist.

### Funding

| Funder | Grant reference number | Author |
| --- | --- | --- |
| Cancer Research UK | A20412 | Christina Ernst<br>Sarah J Aitken<br>Nils Eling<br>Frances Connor<br>Tim F Rayner<br>Margus Lukk<br>Claudia Kutter<br>Duncan T Odom |
| Wellcome Trust | 106563/Z/14/A | Sarah J Aitken |

| European Research Council | 615584 | Duncan T Odom<br>Timothy F Rayner<br>Frances Connor<br>Margus Lukk |
| Wellcome Trust | 098024/Z/11/Z | Robert J Klose |
| European Molecular Biology Laboratory | | Nils Eling |

The funders had no role in study design, data collection and interpretation, or the decision to submit the work for publication.

## Author contributions

CE, Conception and design, Acquisition of data, Analysis and interpretation of data, Drafting or revising the article; JP, HKL, LS, Acquisition of data, Analysis and interpretation of data; SJA, CK, Analysis and interpretation of data, Drafting or revising the article; NE, Analysis and interpretation of data; MCW, FC, TFR, ML, RJK, Contributed unpublished essential data or reagents; DTO, Conception and design, Analysis and interpretation of data, Drafting or revising the article

## Author ORCIDs

Christina Ernst, http://orcid.org/0000-0002-3569-2209
Robert J Klose, http://orcid.org/0000-0002-8726-7888
Claudia Kutter, http://orcid.org/0000-0002-8047-0058
Duncan T Odom, http://orcid.org/0000-0001-6201-5599

## Ethics

Human subjects: Previously published human data from Ward et al. 2013 were used for comparisons in this study.

Animal experimentation: This investigation was approved by the Animal Welfare and Ethics Review Board and followed the Cambridge Institute guidelines for the use of animals in experimental studies under Home Office license PPL 70/7535.

# Additional files

## Major datasets

The following datasets were generated:

| Author(s) | Year | Dataset title | Dataset URL | Database, license, and accessibility information |
|---|---|---|---|---|
| Christina Ernst, Duncan T Odom | 2016 | Chip-Seq analysis of human chromosome 21 after its passage through either the female or male mouse germline | http://www.ebi.ac.uk/arrayexpress/experiments/E-MTAB-4913 | Publicly available at the EBI European Nucleotide Archive (accession no: E-MTAB-4913). |
| Christina Ernst, Duncan T Odom | 2016 | BioCAP-Seq analysis of human chromosome 21 after its passage through either the mouse male germline | http://www.ebi.ac.uk/arrayexpress/experiments/E-MTAB-4930 | Publicly available at the EBI European Nucleotide Archive (accession no: E-MTAB-4930). |
| Christina Ernst, Duncan T Odom | 2016 | RNA-Seq in liver of Tc1 mice carrying human chromosome 21 passaged through either the female or male germline | http://www.ebi.ac.uk/arrayexpress/experiments/E-MTAB-4912 | Publicly available at the EBI European Nucleotide Archive (accession no: E-MTAB-4912). |

The following previously published datasets were used:

| Author(s) | Year | Dataset title | Dataset URL | Database, license, and accessibility information |
|---|---|---|---|---|
| Hannah K Long, Robert J Klose | 2016 | An evolutionarily conserved DNA-encoded logic shapes CpG island formation | http://www.ncbi.nlm.nih.gov/geo/query/acc.cgi?acc=GSE72208 | Publicly available at the NCBI Gene Expression Omnibus (accession no: GSE72208). |
| Ward MC , Wilson MD, Barbosa-Morais NL, Schmidt D, Stark R, Pan Q, Schwalie PC , Menon S, Margus Lukk, Watt S, Thybert D, Claudia Kutter, Kirschner K, Flicek P, Blencowe BJ, Duncan T Odom | 2014 | E-MTAB-1104 - ChIP-seq of human and transgenic mouse adult liver, testes & kidney tissue to investigate epigenetic comparison | http://www.ebi.ac.uk/arrayexpress/experiments/E-MTAB-1104/ | Publicly available at the EBI European Nucleotide Archive (accession no: E-MTAB-1104). |
| Schmidt D, Wilson MD, Ballester B, Schwalie PC, Brown GD, Marshall A, Claudia Kutter, Watt S, Martinez-Jimenes CP, Mackay S, Talianidis I, Flicek P, Duncan T Odom | 2010 | E-TABM-722 - ChIP-seq of Canis familiaris, Gallus gallus, Mus musculus, Homo sapiens, Monodelphis domestica to investigate CEBPA and HNF4a binding in five vertebrates | https://www.ebi.ac.uk/arrayexpress/experiments/E-TABM-722/ | Publicly available at the EBI European Nucleotide Archive (accession no: E-TABM-722). |
| Villar D, Berthelot C, Flicek P , Duncan T Odom | 2015 | E-MTAB-2633 - ChIP-Seq analysis of regulatory evolution in 20 mammals | https://www.ebi.ac.uk/arrayexpress/experiments/E-MTAB-2633/ | Publicly available at the EBI European Nucleotide Archive (accession no: E-MTAB-2633). |

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
