## [Decision Letter]

Thank you for submitting your article "Successful Transmission and Transcriptional Deployment of a Human Chromosome via Mouse Male Meiosis" for consideration by *eLife*. Your article has been favorably evaluated by Marianne Bronner (Senior Editor) and three reviewers, one of whom is a member of our Board of Reviewing Editors. The reviewers have opted to remain anonymous.

The reviewers have discussed the reviews with one another and the Reviewing Editor has drafted this decision to help you prepare a revised submission.

Summary:

In this manuscript, Ernst et al. report successful transmission of the human chromosome 21 derivative Tc1 through the male germ line, albeit at low frequency. This is surprising not only because it demonstrates successful paternal transmission of a trisomic chromosome, but also because it concerns transmission of a human chromosome with a human centromere. In mice, trisomy is associated with germ cell arrest at meiotic metaphase I, due to activation of the meiotic spindle checkpoint by the univalent trisomic chromosome. The spindle checkpoint is more efficient in males than in females, which means that transmission of trisomic chromosomes through the male germ line is problematic. The influence of spermatogenesis versus oogenesis on epigenetic remodelling of trisomic chromosomes has thus been difficult to examine, and the authors present a model, albeit with a human chromosome, where this can be addressed. The authors use this Tc1 mouse model to ask whether paternal versus maternal transmission imparts differences in the transcriptional and epigenetic signatures of the Tc1 chromosome in somatic tissues from offspring. Using a selection of complementary genome-wide approaches to look at chromatin marks, transcription and tissue-specific transcription factor binding in somatic tissues, they find no notable differences, implying that the Tc1 can be equivalently deployed irrespective of parent of origin. The authors also find signs of globally increased gH2AX levels during spermatogenesis in Tc1 males, which they attribute to apoptosis due to aberrant activation of the spindle check point, which may account for the incomplete transmission of the Tc1 chromosome.

This study represents a unique, in depth molecular evaluation of the impact of paternal versus maternal transmission of a trisomic human chromosome in the mouse. It is a succinct, well written and interesting manuscript that will appeal to a broad audience encompassing germ cell and chromosome biologists.

Essential revisions:

1) The authors did not examine the fate of the human chromosome after paternal or maternal inheritance following fertilization and zygotic gene activation. Indeed, any carry over of DNA damage or altered chromatin of the trisomic, human chromosome 21 might lead to loss of some embryos following paternal inheritance. As the authors note that only 11% of offspring, presumably genotyped post weaning, inherit the Tc1 from carrier fathers, it is important to assay when this skewing appears. If at earlier stages of development the% of Tc1 positive offspring is higher, the authors might be assaying a selected population of carriers whose survival is contingent on a particular Tc1 epigenetic signature. The authors could resolve this point by genotyping pre-implantation embryos and/or haploid spermatids from Tc1 fathers. Using superovulated females crossed to the Tc1 males numerous embryos could be collected (morula or blastocyst stages) for genotyping using Tc1 chromosome PCR or FISH. Genotyping spermatids could be performed using Tc1 chromosome FISH.

2) The authors show that in Tc1 males apoptosis, revealed by caspase staining, occurs at meiotic metaphase I (stage XII of the epithelial cycle). To support their conclusion that this causes germ cell arrest, the authors should demonstrate that the numbers of metaphase I cells per tubule in Tc1 males at stage XII are elevated relative to controls (see Odorisio et al. Nat Gen. 1998 18:257 for methodology).

3) The ability to transmit Tc1 through males is surprising, and will be of interest to other groups struggling to perform similar experiments. The authors should comment on how they envisage this property arose in their stock. Is there evidence that the background strains used to maintain their lines exhibit weakened checkpoint mechanisms?

4) Although speculation about human meiosis is not necessarily warranted from this paper, the authors might consider if they can comment on chromosome transmission in people with Down syndrome.

---

## [Author Response]

Essential revisions:

*1) The authors did not examine the fate of the human chromosome after paternal or maternal inheritance following fertilization and zygotic gene activation. Indeed, any carry over of DNA damage or altered chromatin of the trisomic, human chromosome 21 might lead to loss of some embryos following paternal inheritance. As the authors note that only 11% of offspring, presumably genotyped post weaning, inherit the Tc1 from carrier fathers, it is important to assay when this skewing appears. If at earlier stages of development the% of Tc1 positive offspring is higher, the authors might be assaying a selected population of carriers whose survival is contingent on a particular Tc1 epigenetic signature. The authors could resolve this point by genotyping pre-implantation embryos and/or haploid spermatids from Tc1 fathers. Using superovulated females crossed to the Tc1 males numerous embryos could be collected (morula or blastocyst stages) for genotyping using Tc1 chromosome PCR or FISH. Genotyping spermatids could be performed using Tc1 chromosome FISH.*

This was an extremely interesting point that we addressed by characterizing the distribution and fate of the aneuploid human chromosome during spermatogenesis in Tc1 positive males. In sum, we found that a surprisingly high fraction (ca. 95%) of 4N spermatocytes have the aneuploid human chromosome, which is far higher than we (Wilson et al., 2008) or others (O’Doherty et al., 2005) have observed in the past for somatic tissues. Using FISH to genotype 1N spermatids revealed that there was a modest reduction (30% versus expected 50%) in the fraction containing HsChr21. We have added a paragraph (subsection “Efficient passage of a complete human chromosome through mouse male meiosis”, fourth paragraph) and supplemental figure (Figure 3—figure supplement 4) reporting these new results and indicating that this reduction only partially accounts for the much lower Tc1 chromosome transmission rate observed in males versus females. We now elaborate in the Discussion (fifth paragraph) that the balance of ‘missing’ aneuploid mice must disappear either from reduced fertility of the aneuploid 1N sperm or developmental failures of successful fertilizations prior to birth.

*2) The authors show that in Tc1 males apoptosis, revealed by caspase staining, occurs at meiotic metaphase I (stage XII of the epithelial cycle). To support their conclusion that this causes germ cell arrest, the authors should demonstrate that the numbers of metaphase I cells per tubule in Tc1 males at stage XII are elevated relative to controls (see Odorisio et al. Nat Gen. 1998 18:257 for methodology).*

Following this recommendation, we have now performed extensive quantification of the different cell populations in seminiferous tubules of both Tc1 and wild-type mice, resulting in one new figure and one new Results section (see subsection “Tc1 males arrest at metaphase I and display chromosome congression defects” and Figure 2 with figure supplements 1 and 2).

We observe a consistent increase in the number of Stage XII tubules, as well as 4N spermatocytes and meiotic cells in Tc1 tubules compared to wild-types. This is consistent with an arrest at metaphase I, and a subsequent reduction of both round and elongating spermatids in Tc1 males, which we observed both in our immuno-fluorescence stainings as well as with FACS.

The manual classification of pro-metaphase and metaphase cells did not yield as strong results as described in Odorisio et al., however, we did observe a clear trend towards an increase of metaphase I cells in 3 out of the 5 inspected Tc1 males. The mice investigated by Odorisio et al. were completely sterile due to their arrest at metaphase I; in contrast, most Tc1 males were fertile and were thus not expected to show as strong a block at metaphase I. Interestingly, these experiments also revealed that congression defects occur more frequently in Tc1 tubules compared to wild-types. These lagging chromosomes would likely trigger the spindle assembly checkpoint, causing the block at metaphase I and thus increased apoptosis of Stage XII tubules.

*3) The ability to transmit Tc1 through males is surprising, and will be of interest to other groups struggling to perform similar experiments. The authors should comment on how they envisage this property arose in their stock. Is there evidence that the background strains used to maintain their lines exhibit weakened checkpoint mechanisms?*

Our inquiries and literature searches have not identified published studies that directly address this issue. We ourselves have never attempted breeding setups in other genetic backgrounds, but have now added a section to the Discussion describing the breeding experiments performed by O’Doherty et al. (see Discussion, sixth paragraph).

We have compared total testis RNAseq data from wildtype BL6 mice as well as our 129S8B6 F1 mice that we use for breeding with Tc1 animals to see whether there are any changes on the transcriptional level that could explain a weakened spindle checkpoint (Discussion, sixth paragraph). We observed only moderate changes between these two strains and GO analysis did not identify any mechanisms involved in meiotic control mechanisms. Amongst the spindle checkpoint genes we observed Septin2 to be differentially expressed, but only to a very modest extent (see Figure 7). While this could be an interesting candidate for future investigations given its previous implications in the control of metaphase-to-anaphase transition in female oocytes (Zhu et al., 2010), cell cycle control mechanisms usually go beyond transcriptional control. Therefore, little conclusions can be drawn from an increased expression level of this gene, as the crucial regulatory mechanisms could be post-translational.

Author response image 1.**DOI:**
http://dx.doi.org/10.7554/eLife.20235.018

*4) Although speculation about human meiosis is not necessarily warranted from this paper, the authors might consider if they can comment on chromosome transmission in people with Down syndrome.*

We have added a section in the Discussion comparing our gender ratios with the observed sex bias in human DS. While there is an increase in males with human DS, especially when the origin is paternal, the underlying mechanism has not been demonstrated. We do not observe such a bias in neither female or male germline transmission of the aneuploid chromosome and can therefore only state that the Tc1 model is not suitable to dissect the underlying mechanisms of the sex bias observed in human DS (see Discussion, seventh paragraph).